# Spinning a Sustainable Future: Electrospun Polysaccharide–Protein Fibers for Plant-Based Meat Innovation

**DOI:** 10.3390/foods13182962

**Published:** 2024-09-19

**Authors:** Letícia G. da Trindade, Letícia Zanchet, Fabiana Perrechil Bonsanto, Anna Rafaela Cavalcante Braga

**Affiliations:** 1Department of Chemical Engineering, Universidade Federal de São Paulo (UNIFESP), Diadema 09913-030, Brazil; lgt.gtrindade@gmail.com (L.G.d.T.); fabiana.perrechil@unifesp.br (F.P.B.); 2LRC-Institute of Chemistry, Universidade Federal do Rio Grande do Sul (UFRGS), Porto Alegre 91501-970, Brazil; leticia_zanchet@hotmail.com; 3Nutrition and Food Service Research Center, Universidade Federal de São Paulo (UNIFESP), Santos 11015-020, Brazil

**Keywords:** electrospinning, meat analogs, texture

## Abstract

This study aims to evaluate the feasibility of producing electrospun fibers by combining polysaccharides, zein, and poly(ethylene oxide) (PEO) to simulate the fibers applied in plant-based meat analogs. The rheological properties of biopolymer solutions were evaluated, and the electrospun fibers were characterized according to their morphology, structural interactions, and thermal analysis. The results indicated that the fibers prepared in a ratio of 90:10 of zein/carrageenan from the mixture of a solution containing 23 wt.% of zein with a solution containing 1 wt.% of carrageenan and with the addition of 1 wt.% of PEO presented a promising structure for application as fibers in meat analogs because they have a more hydrophilic surface. Thus, they have good moisture retention. In addition, they have good thermal stability at high temperatures, which is crucial to achieve a consistent and pleasant texture. Furthermore, it was observed that adding zein and PEO helps with the spinnability of the polysaccharides, producing fibers with good homogeneity.

## 1. Introduction

Polysaccharides are complex carbohydrate molecules found in foods mainly of plant origin and are an essential source of energy for the human body as they perform several vital functions for health [1]. One of the main functions of polysaccharides in human nutrition is to provide energy gradually and sustainably. Unlike simple carbohydrates, such as refined sugars, which are quickly metabolized, polysaccharides give a slower release of glucose, preventing blood sugar spikes and keeping energy levels stable throughout the day. Furthermore, polysaccharides also play an essential role in preventing chronic diseases such as type 2 diabetes, cardiovascular diseases, and some types of cancer [2].

The polysaccharides that have been gaining prominence in the food industry are alginate, carrageenan, and pectin. Alginate is an anionic linear polysaccharide existent in many species of brown seaweed. Its chemical structure consists of (1–4) linked β-D-mannuronic acid and α-L-glucuronic acid units in various sequences and compositions [3]. Sodium alginate is a polyelectrolyte with high conductivity and surface tension, making electrospinning of this biopolymer challenging. Even though alginate can form solutions with a wide range of viscosity, it has a repulsive force among the polyanions, acting as a barrier. This repulsive force is the key factor hindering the electrospinning of sodium alginate [4]. An alternative to reduce these repulsive forces is the interactions, for example, with PEO that can result in successful electrospinning of sodium alginate/PEO blends [5,6].

Incorporating small amounts of PEO is a viable option for enhancing the elasticity of fiber formations that include zein and other proteins. PEO is a hydrophilic polymer widely regarded as safe for food applications due to its non-toxic nature, biocompatibility, and biodegradability. Furthermore, among the polymers discussed in the existing literature, PEO stands out for its biodegradability and classification as Generally Recognized as Safe (GRAS) by the Food and Drug Administration (FDA UNII 16P9295IIL) [7,8]. Consequently, it can be confidently utilized in processed foods and beverages.

Carrageenan is a polysaccharide with gelation properties and mechanical strength and has an innate resemblance to natural glycosaminoglycans present in the native extracellular matrix [9]. On the other hand, pectin is one of the most complex heterogeneous polysaccharides in nature, is composed of mainly (98%) linear chains of (1,4)-linked-α-d-galacturonic acid, and also contains other pectic domains (rhamnogalacturonan-I and rhamnogalacturonan-II) [10,11,12]. This polysaccharide is widely utilized as a gelling agent in the food technology and processing industries. However, the inconsistencies in its chemical structure due to commercial extraction methods and intrinsic source variability have limited its use in other applications [13].

In recent years, polysaccharides have gained prominence in biotechnology due to their unique properties, but these gels are viscous and difficult to handle, which makes the electrospinning process difficult. To produce fibers by electrospinning, it is best to use modified polysaccharides with a more organized structure and greater rigidity, ensuring that the fibers formed are high quality and suitable for various applications. These electrospun polysaccharide fibers have aroused interest in several areas, such as medicine, the textile industry, and tissue engineering, and show great potential in food science electrospinning [14].

In the food industry, these fibers can be used as food additives to improve a food’s texture, stability, and water absorption, among other benefits. These fibers possess unique properties due to their extremely small size, high surface area, and porous structure. These properties can enable the controlled release of bioactive compounds into foods, improve adhesion on food surfaces, and provide additional health benefits, such as reducing the absorption of fats or sugars. However, electrospinning of polysaccharides is challenging, since some properties of polysaccharide solutions are not optimal for electrospinning. It is necessary to overcome challenges such as the high conductivity of polyelectrolyte solutions [15] and the high surface tension of their aqueous solutions [16].

Stijnman et al. [17] evaluated the reliability of a variety of polysaccharides commonly used in the food industry, and the results showed that solutions of high-methoxyl pectin (3.4%), alginate (4%), κ-carrageenan (1%), and λ-carrageenan (1%) are not able to form fibers. Neither variation of the electrospinning conditions (electric field and distance to the collector and flow rate) influenced the ability to form fibers within the settings tested. If the polysaccharide solution could not form fibers or a jet at the standard settings, varying the field, distance, or flow rate made no difference.

On the other hand, our research group has successfully developed electrospun fibers with various polymers, including PEO, zein, alginate, gelatin, and a combination of them [7,8,18,19,20].

Various approaches can be used to overcome these drawbacks: (i) the addition of carrier polymers that facilitate the electrospinning of polysaccharide solutions, (ii) the insertion of additives that increase the spinnability, (iii) the introduction of surfactants, (iv) the use of cosolvents, and (v) the chemical modification of polymers to increase their solubility and spinnability [21].

This research is focused on developing electrospun fibers of alginate, carrageenan, and two types of pectin, combined with zein and PEO, for future application in plant-based meat analogs.

Zein was chosen because it is a protein found in corn and is becoming one of the primary raw materials used in electrospinning [22]. Zein is often used in the food industry for its properties of encapsulation, solubility, and biodegradability [23]. Zein, in electrospinning, has shown great potential as a versatile and sustainable alternative for producing ultrafine fibers, and it can also be used as an additive in polymers, improving their mechanical properties, such as tensile strength and toughness [24]. Its unique properties, such as biodegradability and encapsulation, make it an innovative material for diverse applications, from tissue engineering to developing advanced materials.

The combination of polysaccharides with zein not only enables the formation of fibers by electrospinning but can also result in fibers with an improved texture. Texture plays a crucial role in the quality and acceptance of plant-based foods, being a complex physicochemical and sensory attribute that depends on the materials’ mechanical, fracture, and surface properties [25]. Plant polysaccharides are widely used as functional food ingredients due to their thickening, gelling, structuring, binding, and fluid retention properties [26]. Thus, the interactions between polysaccharides and proteins can be exploited to create new or improved food textures [27].

The PEO addition to the polysaccharide/zein blend was chosen to give more elasticity to the fibers. PEO is commonly employed as a representative model in electrospinning procedures due to its water solubility, ease of electrospinning, and non-toxic nature [28].

## 2. Materials and Methods

### 2.1. Materials and Solution Preparation

Sodium alginate (GRINDSTED^®^ FD175) was obtained from Danisco (Copenhagen, Denmark), κ-carrageenan was obtained from FMC BioPolymer (Philadelphia, PA, USA), high-methoxyl (HM) pectin (type USP-B; degree of esterification ~60%) was kindly supplied by CPKelco (Limeira, SP, Brazil) and low-methoxyl (LM) pectin (degree of esterification = 18.5%) was obtained from GastronomyLab (Brasília, DF, Brazil). All polysaccharides were used as received. Zein, ethanol (99.8%), and poly(ethylene oxide) (PEO) (900,000 g.moL^−1^) were provided by Sigma-Aldrich (St. Louis, Missouri, USA).

#### 2.1.1. Alginate/PEO Solutions

Alginate was dissolved in deionized water at a concentration of 2.0% *w*/*v*. PEO was dissolved in deionized water at a concentration of 4.0% *w*/*v*. We prepared three solutions with different volume ratios (alginate/PEO: 20:80, 40:60, and 50:50). These solutions were mixed overnight at room temperature with constant stirring.

To prepare the alginate/zein/PEO solutions, the 23 wt.% zein was first dissolved in an 80% ETOH solution and stirred until its complete dissolution, which took approximately 1 h. The concentration of the alginate solution was reduced from 2 wt.% to 1 wt.% to decrease the viscosity of the solution. With the two solutions ready, zein was added to the predefined amount of alginate (zein/alginate: 80:20, 85:15, and 90:10). After one hour under constant agitation, PEO was slowly added to the mixture. Two concentrations of PEO, 0.3 and 1 wt.%, were evaluated.

#### 2.1.2. Zein/Carrageenan/PEO Solutions

Polymer zein/carrageenan/PEO solutions were prepared in the following ratios: 80:20, 85:15, and 90:10 with 0.3 or 1.0 wt.% of PEO. The polymeric solutions were prepared separately. First, a 1 wt.% solution was prepared by dissolving carrageenan in deionized water; this solution was maintained under constant stirring for 1 h at room temperature for the complete dissolution of the carrageenan. Simultaneously, a 23 wt.% zein solution in 80% ethanol was prepared and stirred for 1 h. After this time, the solution of zein and carrageenan was added at the predetermined concentrations, and the stirring was maintained for another 1 h. At the end of this time, 0.3 or 1 wt.% of PEO was slowly added, and stirring was continued for another 1 h.

#### 2.1.3. Zein/Pectin/PEO Solutions

For the electrospun natural pectin fibers, we used two different pectins: HM pectin and LM pectin. The zein/pectin/PEO polymer solutions were prepared in the following ratios: 80:20, 85:15, and 90:10 with 0.3 or 1.0% PEO (*w*/*w*). The polymeric solutions were prepared separately. First, a 1% (*w*/*w*) solution was prepared by dissolving pectin in deionized water; this solution was maintained under constant stirring for 1 h at room temperature for the complete dissolution of the pectin. Concomitantly, a solution of 23% (*w*/*w*) zein in 80% ethanol was prepared and kept under stirring for 1 h. After this time, the solution of zein and pectin was added at the predetermined concentrations, and the stirring was maintained for another 1 h. At the end of this time, 0.3 or 1% PEO (*w*/*w*) was slowly added, and stirring was continued for another hour.

Samples containing alginate, carrageenan, HM pectin, and LM pectin were named, respectively, as “Z-A(X:Y)/PW”, “Z-C(X:Y)/PW”, “Z-PHM(X:Y)/PW”, and “Z-PLM(X:Y)/PW”, in which X is the proportion of zein solution (80, 85, or 90), Y is the proportion of polysaccharide solution (20, 15, or 10), and W is the PEO concentration (0.3 or 1).

### 2.2. Solution Characterization

Rheological characterizations were performed on an MCR 92 rheometer (Anton Paar, Austria) at 25 ± 0.2 °C. Each sample was loaded under a 25 mm or 50 mm (depending on the solution’s viscosity) parallel plate, and a 1 mm gap was applied at the shear rate from 0.1 s^−1^ to 300 s^−1^. A three-step (up–down–up) program was used, and the third curve was considered the steady state. The measurements were carried out in triplicate.

Rheological parameters were obtained by fitting the Power Law model (Equation (1)) to the flow curves at the steady state:(1)$$\sigma =k\cdot {\dot{\gamma }}^{n}$$
where *σ* is the shear stress (Pa), *k* is the consistency index (Pa·s^n^), *n* is the flow behavior index (−), and $\dot{\gamma }$ is the shear rate (s^−1^).

### 2.3. Preparation of Polysaccharides/Protein Fibers

The polysaccharide/protein solutions were loaded into a 5 mL syringe attached to a tip of 1.03 mm internal diameter, and electrospinning was conducted using a laboratory-scale electrospinning machine (FLUIDNATEK LE-10, BIOINICIA, Paterna, Valencia, Spain). The fibers were collected in a rotatory collector covered with aluminum foil for easy removal. The polymer solution was pumped through the syringe at 1800–3000 µL/h. The distance from the end of the needle to the collector plate was fixed at 12–15 cm. The voltage was varied from 18 to 24 kV. All nanofibers were made at room temperature (20–25 °C) and 50–60% relative humidity.

### 2.4. Characterization of Electrospun Fibers

The morphology of the nanofibers was examined using a scanning electron microscope (SEM) (JEOL LV 6600, Peabody, MA, USA). The size distribution of fiber diameters was determined using ImageJ software (13.0.6.). For this analysis, 100 randomly selected points were chosen from the SEM images. To investigate the structural interactions in the samples, the ATR-FTIR spectra of the nanofibers were obtained using a Nicolet 6700 model (Madison, WI, USA) equipped with a germanium crystal in ATR mode. The spectra were collected in the wavenumber range of 4000–650 cm^−1^ with 128 scans. The thermal stability of the fibers was assessed through TGA (Discovery-TA equipment, New Castle, DE, USA) with a flow of ultra-pure nitrogen at 60 mL min^−1^ as the purge gas. The decomposition of the fibers was analyzed between 25 and 700 °C, using a heating rate of 10 °C min^−1^. The hydrophilicity of the fibers was determined by measuring the contact angle of a water drop placed on the sample using a DataPhysics OCA 11 goniometer (Filderstadt, Baden-Württemberg, Germany) at 25 °C.

### 2.5. Statistical Analyses

The experiments were performed in triplicate and expressed in terms of the mean and ± standard deviation. Additionally, the measurements from the samples, using the MEV images, were carried out independently. In both cases, the samples were compared by applying an analysis of variance (ANOVA) using the degree of significance of 95% (*p* < 0.05), followed by Tukey’s test.

## 3. Results

### 3.1. Zein/Alginate/PEO

A list of alginate and alginate/PEO solutions tested under different proportions and conditions is shown in Table 1. To verify the reliability of the polysaccharide solution, a 2 wt.% solution was tested with different parameters.

Despite changing the electrospinning parameters, no fibers were formed. Then, alginate/PEO solutions were made to give the possible fiber formed more elasticity. The ratios studied were 20:80, 40:60, and 50:50 of solutions of 2 wt.% of polysaccharide and 1 wt.% of PEO.

The data show that the alginate/PEO blends cannot form a jet. These solutions cannot form the Taylor cone; they form drops at the tip of the needle that fall. Modifying the electrospinning conditions, electric field, and flow rate did not influence the ability of fiber formation in the tested samples.

With the impossibility of forming fibers using pure alginate and the alginate/PEO blend, zein was incorporated into the mixture. Solutions of 80:20, 85:15, and 90:10 of zein/alginate were prepared. To these solutions, 0.3 wt.% or 1 wt.% of PEO was added. We fixed the feeding rate for these tests at 2000 μL/h and varied the voltage and TCD. Table 2 shows the composition of the samples, their designation, the parameters used, and the microscopic observation of the collector.

The data in Table 2 show that a jet is formed with these parameters and the addition of zein. However, drops accompany them, and fibers are not formed because the jet cannot reach the collector. By adjusting two parameters of the electrospinning conditions, voltage, and TCD, the solution manages to create a jet that reaches the collector, where the formation of fibers is observed. However, there is no fiber formation for polymeric solutions with higher alginate concentrations (zein/alginate ratios of 85:15 and 80:20) with the addition of 0.3% PEO. For a higher concentration of PEO (1%), an increase in the alginate concentration in the solutions leads to the formation of fibers. Still, a decrease in the quantity of formed fibers is observed.

The morphology of the electrospun fibers is influenced by the nature of the solution used during the electrospinning process. The solution’s molecular chain structure and behavior greatly impact its properties, affecting the electrospun fiber’s appearance.

Figure 1a shows the relationship between shear rate and viscosity for the polymer solutions, and the rheological parameters are summarized in Table 3.

As can be seen, the polymeric solutions have a shear-thinning behavior. This is confirmed by the flow index (n) values obtained (Table 3), which are less than 1, which indicates that the polymer solutions show pseudoplastic behavior. The obtained coefficient (R^2^) ranged between 99.802 and 99.992, confirming that the model used well-fitted flow curves. The consistency coefficient (K) increases with the addition of alginate for both the solution containing 0.3 and the solution containing 1% PEO, indicating an increase in the apparent viscosity.

The zein/PEO spectra showed a band at 3292 cm^−1^ that refers to -NH_2_ stretching in zein and the bands that were attributed to amide I, II, and III at 1650 (C=O stretching), 1534 (N-H bending and C-N stretching), and 1240 cm^−1^, respectively [29]. The band observed at 1091 cm^−1^ refers to the C-O-C ether linkage of PEO [30]. When zein and alginate are mixed in different proportions with PEO, the spectra are similar regardless of the proportion used. The absorption band at 2870 cm^−1^ observed in the PEO powder sample attributed to the C-H stretching symmetric and asymmetric is not so evident for any of the samples [31]. The characteristics bands of sodium alginate typically appear at 3410 cm^−1^ (-OH), 1635 cm^−1^ (asymmetric stretching vibration of COO groups), 1419 cm^−1^ (symmetric stretching vibration of COO groups), and 1050 cm^−1^ (elongation of C-O groups) [32,33]. However, in this case, as the characteristic bands of zein appear in almost the same regions, it was impossible to distinguish them.

The ATR-FTIR spectra of zein, PEO, and alginate powders, as well as zein/PEO and zein/alginate/PEO fibers, are illustrated in Figure 2. Figure 3 shows the SEM images and fiber diameters obtained by varying the PEO concentration and zein/alginate ratio. Adding alginate to the zein/P03 solution in a ratio of alginate/zein of 90:10 reduces the fibers’ diameter from 6.01 µm to 3.95 µm. However, it can be observed that more homogeneous fibers are formed in a more significant number, as shown in Figure 3a,b. Increasing the PEO concentration from 0.3 to 1% reduces the fiber diameter from 6.01 µm to 4.34 µm.

However, more homogeneous fibers are formed, as shown in Figure 3a,c. It can be seen that compared to the Z23P1 sample (Figure 3a), there is a slight reduction in the fiber diameter for the Z-A (90:10)/P1 sample (Figure 3d), and there is a reduction in the homogeneity of these fibers. When the alginate ratio increases to 85:15 (*w*/*w*) (Figure 3e), the fiber diameter is reduced significantly, approximately 33%. In addition, it can be observed the formation of some fibers occurs with diameters between 8 and 10 µm. On the other hand, when the alginate concentration in the zein/alginate ratio is increased to 80:20 (*w*/*w*) (Figure 3f), the fiber diameters increase again.

The hydrophobic characteristics of a material are reflected by the contact angle of water on its surface, which depends on the material’s molecular structure and surface morphology [34]. Figure 4 shows the water contact angles for the electrospun fibers.

It can be seen that the contact angle decreases from 97.3 ± 2.2 to 88.4 ± 1.3 with an increasing PEO concentration, indicating that there is an increase in the hydrophilicity of the fibers. This behavior can be attributed to PEO being a hydrophilic polymer [31]. The zein/alginate blend at a ratio of 90:10 with the addition of 0.3 wt.% PEO, the Z:A (90:10)/P03 sample, forms fibers with a slightly more hydrophobic surface (104.1 ± 1.4) than the sample that does not contain alginate (97.3 ± 2.2), the Z23P03 sample. On the other hand, increasing the alginate concentration in the samples with 1 wt.% PEO leads to a decrease in the contact angle from 97.4 ± 1.5 (Z:A (90:10)/P1 sample) to 78.2 ± 2.0 (Z:A (80:20)/P1 sample), which means that increasing the alginate concentration increases the hydrophilicity of the zein/alginate/PEO fibers. This behavior can also be related to the fiber diameter (Figure 3), as fibers with smaller diameters tend to be more competitive in terms of surface area, which can result in smaller contact angles and, therefore, better wetting properties. For the samples Z:A (90:10)/P1 and Z:A (80:20)/P1, the fiber diameter reduces from 3.94 ± 1.94 μm to 3.20 ± 1.99 μm, respectively.

The thermal stability of electrospun zein fibers with the addition of PEO and alginate in different proportions was evaluated using thermogravimetric analysis (TGA), as shown in Figure 5. For all the studied formulations, it was observed that the first thermal event occurred between 30 and 200 °C, indicating the evaporation of the free water present in the films and powders (a) and (b).

The second region of the graph indicates the onset of thermal degradation. It was found that in this region, the degradation temperature of zein powder is at 270 °C, according to the literature [35]. In the case of pure PEO powder, the degradation temperature is around 320 °C [36]. The influence of PEO in the zein matrix can be observed through the increase in the degradation temperature observed in the film with 3% PEO (Z23PO3) (a), where it reaches 280 °C. On the other hand, in the fiber with 1% zein (Z23P1) (b), the degradation temperature is within the range of pure zein. In both proportions of PEO in the zein matrix, the insertion of PEO does not negatively influence the degradation temperature of the films.

Regarding the alginate powder, a sharp drop was observed in the first stage, indicating the presence of water in its matrix. The second thermal event was attributed to the decomposition of the sodium constituent of alginate [37].

All blends with the addition of different proportions of alginate did not show significant differences in their degradation temperatures, indicating that the zein matrix with these different proportions has thermal stability in the range of 270–280 °C. This suggests that adding alginate contributes to the integrity of the fibers at higher temperatures. Despite low alginate concentrations, it is confirmed that its incorporation into zein/PEO blends maintains its thermal stability.

### 3.2. Zein/Carrageenan/PEO

With fibers successfully obtained for the studied ratios of the zein/alginate/PEO blends, solutions were made with the same ratios of zein/carrageenan/PEO to evaluate the reliability of this mixture. The macroscopic and microscopic observations and the Power Law parameters for zein/PEO and zein/carrageenan/PEO solutions are presented in Table 4.

The data presented in Table 4 show that the mixture with biopolymer/polymer improves carrageenan processing, since a jet forms in all evaluated conditions.

We observed the formation of fibers with good spinnability/playability, except for when using the ratios of zein/carrageenan of 80:20 and 85:15. However, it is also observed that increasing the concentration of polysaccharides in the mixture decreases the number of fibers formed. The reduction in the number of fibers formed can be influenced by factors such as (I) the solution viscosity: polysaccharides can increase the viscosity of the solution, which can help to stabilize the polymer jet during electrospinning; a more viscous solution tends to produce more continuous and uniform strands, reducing the formation of multiple fibers or thinner “threads”; (II) the interaction between zein and the polysaccharide: the chemical or physical interaction between zein and polysaccharides can result in a more homogeneous polymer matrix, which can improve the alignment of molecules during the electrospinning process; this can result in fewer breakages and, therefore, fewer fibers; (III) fiber size increase: the introduction of polysaccharides can increase the diameter of the fibers formed due to their presence in the solution; thicker fibers may be less likely to split during the process, thus reducing the total number of fibers produced; (IV) elastic and mechanical properties: polysaccharides can provide mechanical properties that help stabilize the fiber during solidification, preventing excessive tufting or agglomeration; and (V) evaporation rate control: the polysaccharide can influence the solvent evaporation rate, affecting the capture of the polymer jet and, consequently, fiber formation. More controlled evaporation can result in more uniform fiber production [17,38,39]. Combined, these factors can lead to fewer fibers forming during electrospinning, resulting in a final material with specific characteristics.

In Figure 6a, the connection between shear rate and viscosity for the Z23P03, Z23P1, Z-C(90:10)/P03, and Z-C(90:10)/P1 polymer solutions is depicted, while the rheological parameters are summarized in Table 4. Increasing the PEO concentration increases the viscosity of the zein/PEO polymeric solution. When carrageenan is added to these solutions, the viscosity increases for the solution with 0.3% PEO, and a reduction in the viscosity for the solution with 1% PEO occurs.

From the parameters in Table 4, the coefficient (R^2^) ranged between 99.931 and 99.992, confirming that the Power Law model used well-fitted flow curves. The flow index (n) is less than 1, indicating that the polymer solutions showed a pseudoplastic behavior, as confirmed in Figure 6b. The n reduces with the addition of carrageenan in the zein/PEO solutions, indicating an increase in the pseudoplasticity of the solutions. The consistency coefficient (K) increases with the addition of carrageenan to the zein/PEO03 solution, indicating an increase in the apparent viscosity. However, an opposite behavior is observed when carrageenan is added to the zein/PEO1 solution.

The ATR-FTIR spectra of zein (23% (*w*/*w*))/PEO (1% (*w*/*w*)) and zein/carrageenan for different ratios are illustrated in Figure 7. The zein/PEO spectra showed a band at 3292 cm^−1^ that refers to -NH_2_ stretching in zein as well as bands attributed to amide I, II, and III at 1650 (C=O stretching), 1534 (N-H bending and C-N stretching), and 1240 cm^−1^, respectively [29]. The band observed at 1091 cm^−1^ refers to the C-O-C ether linkage of PEO [40]. The spectra of the electrospun fibers composed of zein and carrageenan in different proportions containing 1% PEO are similar to the spectrum of zein/PEO. In these spectra, it is not possible to identify the bands at 3319 cm^−1^ (OH stretching vibration), 1618 cm^−1^ (H–O–H deformation band), 1428 cm^−1^ (CH_2_ in-plane bending), 1234 cm^−1^ (Sulfate stretching of S–O), 1159 cm^−1^ (C–O stretching), 920 cm^−1^ (C–O–C of 3,6-anhydro galactose), 844 cm^−1^ (C–H rocking), and 699 cm^−1^ (sulfate on C-4 galactose), which are characteristic of carrageenan [41]. Observing these bands is impossible because the distinctive bands of zein and carrageenan appear in the same regions.

Figure 8 shows the SEM images and the fiber diameters obtained by varying the zein/carrageenan ratio and the PEO concentration. For the Z23P03 sample, Figure 8a, it can be observed that inhomogeneous fibers are formed with a diameter of 6.01 µm. When the zein/carrageenan ratio is 90:10, more homogeneous fibers are formed, and the fiber diameter increases by approximately 1.48 times (8.89 µm). When the PEO concentration increases from 0.3 to 1%, there is a reduction in the diameter of the fibers, but they become more homogeneous, as shown in Figure 8a,c. On the other hand, when there is a variation in the carrageenan/zein ratio, we can observe that a lower concentration of carrageenan in the solution leads to larger and more homogeneous fibers, as shown in Figure 8d–f. This behavior can be explained by changes in the viscosity of the solution (Table 4). An increase in the concentration of PEO in the zein/carrageenan samples decreases the viscosity of the solution. This reduction in viscosity makes the solution more fluid, which makes the stretching force applied during electrospinning more effective. This stretching force is responsible for stretching the solution from the needle, forming thinner fibers.

On the other hand, an increase in the zein concentration decreases the viscosity of the polymer solution, resulting in larger diameter fibers. This increase in the fiber diameter with decreasing solution viscosity may have been influenced by factors such as solution flow, surface tension, stretchability, polymer concentration, and jet formation. These factors interact in complex ways and may vary depending on the type of polymer, solution composition, and experimental conditions, such as the voltage, distance between the electrofusion source and the collector, and solution flow rate. It has been reported that the electrospinnability was significantly improved by adding zein, which was attributed to the fact that zein particles can act as plasticizers to reduce hydrogen bond interactions [42].

The surface hydrophobicity of nanofiber samples was investigated by measuring the static water contact angle. Figure 9 illustrates that the water contact angle value of Z23P03 nanofiber was 97.3°, which decreased to 88.4° with an increase in the PEO concentration (Z23P1 sample). Despite zein being generally hydrophobic, hydrophilic amino acids are in its protein structure [42]. These hydrophilic amino acids allow water molecules to penetrate the fiber and reach the hydrophilic PEO layer in the fiber core, increasing the overall hydrophilicity. Thus, the surface of the fiber becomes more hydrophilic as the amount of PEO increases. When carrageenan is added, there is a reduction in the contact angle, and the higher the concentration of carrageenan in the fibers, the lower the contact angle. This fact can be explained by carrageenan having a significant content of hydroxyl groups and being entirely negatively charged, as it is a hydrophilic biomaterial [43].

Thermogravimetric analysis revealed two distinct degradation stages for all samples, as shown in Figure 10. The first degradation occurred in the 30–200 °C temperature range and was attributed to the evaporation of residual water in the materials (a) and (b). In the case of carrageenan powder, the weight loss between 30 °C and 200 °C represented the evolution of light volatiles, along with additional removal of bound residual water. A second mass loss occurred between 200 °C and 230 °C, revealing the devolatilization of thermally more stable volatile compounds in carrageenan. Around 230 °C, the highest rate of weight loss corresponded to the decomposition of carbonaceous materials. The carrageenan-modified fiber Z-C(90:10)/P03 exhibited thermal stability comparable to the Z23P03 fiber. Both samples showed mass loss at 280 °C in the second degradation stage. The same behavior was observed in fibers that contained 1% PEO in the composition (Z23P1, Z-C(80:20)/P1, Z-C(85:15)/P1, and Z-C(90:10)/P1). This favorable behavior demonstrates that adding carrageenan does not interfere with the thermal stability of the fibers [44].

The same proportions of zein/polysaccharide/PEO mixtures used for alginate and carrageenan were used for HM and LM pectins. Table 5 presents the macroscopic and microscopic observations and the Power Law parameters for the solutions.

The data presented in Table 5 show that the mixture with biopolymer/polymer improves the processing of pectin, since in all evaluated conditions, a jet forms. There is the formation of fibers with good spinnability/playability, except for when using the ratios of zein/pectin of 80:20 and 85:15 with 0.3% PEO. However, it is also observed that increasing the concentration of pectin in the mixture decreases the quantity of fibers formed.

The flow curves of the zein/PEO and zein/pectin/PEO solutions are shown in Figure 11, and the Power Law parameters are shown in Table 5. The viscosity vs. shear rate curves reveal that the viscosity values increase with the addition of both evaluated pectins, as shown in Figure 11a,b. The LM pectin increases the viscosity of the solution with a lower PEO concentration (Figure 11a). The HM pectin increases the viscosity of the solution with a higher PEO concentration (Figure 11b).

The coefficient (R^2^), shown in Table 5, ranged between 86.13 and 99.99, confirming that the Power Law model fitted the flow curves. Figure 11c,d confirms that the polymer solutions exhibited a pseudoplastic behavior, as indicated by a flow index (n) value below 1 (Table 5). The addition of pectin to both the solution containing 0.3% and that containing 1% PEO results in an increase in the consistency coefficient (K), which suggests an elevation in the apparent viscosity.

### 3.3. Fibers Characterization

The electrospun fiber spectra of the zein (23% (*w*/*w*))/PEO (1% (*w*/*w*)) blend and samples containing different proportions of LM or HM pectin with different PEO amounts are shown in Figure 12. The zein/PEO spectra showed a band at 3292 cm^−1^ that refers to -NH_2_ stretching in zein, as well as bands attributed to amide I, II, and III at 1650 (C=O stretching), 1534 (N-H bending and C-N stretching), and 1240 cm^−1^, respectively [29]. The band observed at 1091 cm^−1^ refers to the C-O-C ether linkage of PEO [35]. The spectra of the electrospun fibers composed of zein and LM pectin in different proportions containing 1% PEO, shown in Figure 12a, are similar to the spectrum of zein/PEO. When HM pectin, shown in Figure 12b, replaces LM pectin, it is also impossible to distinguish the characteristic bands of pectin. These bands should be observed at 3418 cm^−1^ (O-H stretching vibration), 2936 cm^−1^ (sp^3^ C-H), 1630.49 cm^−1^ (C=O stretching vibration), 1439 cm^−1^ (sp^3^ CH_2_ of methylene bridge), 1058 cm^−1^ (C-O stretching vibration), and 770 cm^−1^ (sp^2^ C-H bending vibrations) [45].

Figure 13 presents the SEM images and the fiber diameters obtained by varying the LM pectin/zein ratio and the PEO concentration in the samples. The addition of LM pectin to the solution containing zein/PEO03 leads to a reduction in the fiber diameter. However, an increased quantity and homogeneity of the fibers formed can be observed (Figure 13a,b). Adding LM pectin to zein/PEO1 increases the fiber diameter; an almost double fiber diameter increase is observed when varying the LM pectin/zein ratio, as shown in Figure 13c–f. The increase in the pectin amount in this ratio (pectin/zein) is responsible for the increase in the fiber diameter. However, when increasing the amount of pectin, we can observe that the fibers formed are less homogeneous.

Figure 14 presents the SEM images and fiber diameters obtained by varying the HM pectin/zein ratio and PEO concentration in the samples. Essentially, adding HM pectin to the solution containing zein/PEO03 does not change the diameter of the fibers. However, it can be observed that there is greater homogeneity of the fibers formed, as shown in Figure 14a,b. Adding HM pectin increases the fiber diameter by 2.11 times compared to the sample with zein/PEO1, as shown in Figure 14c,d. However, when we look at the different HM pectin/zein ratios studied, 85:15 and 80:20, we see that increasing the amount of pectin and reducing the amount of zein reduces the fibers to a 1.38 µm diameter when the ratio is 80:20. The formation of non-uniform fibers occurs.

Figure 15 presents the results of contact angle measurements for zein/PEO and zein/PEO electrospun fibers containing the different pectins studied at different zein/pectin ratios. The contact angle is reduced with the pectin addition in the proportion of 90:10 (zein/pectin) in the samples with different PEO concentrations, and there is a reduction of 1.5% in the samples with 0.3 wt.% PEO (97.3 ± 2.2 to 64.8 ± 1.5 (Z-PLM (90:10)/P03) and of 1.2% in the samples with 1 wt.% PEO (88.4 ± 1.3 to 73.3 ± 1.8 (Z-PLM (90:10)/P1), as shown in Figure 15a. The increase in the concentration of LM pectin in the samples with 1 wt.% PEO leads to a reduction in the contact angle. It reduces from 73.3 ± 1.8 (Z-PLM (90:10)/P1) to 15.22 ± 1.5 (Z-PLM (80:20)/P1). This same behavior is observed for the samples with HM pectin, and the contact angle reduces from 102.4 ± 1.8 (Z-PHM (90:10)/P1) to 71.1 ± 2.0 (Z-PHM (80:20)/P1), as shown in Figure 15b. This behavior can be explained by extra hydrogen bonding between water and the hydroxyl groups of the incorporated pectin [41]. Generally, fibers with smaller diameters tend to be more competitive in terms of surface area, which can result in smaller contact angles and, therefore, better wetting properties. However, samples containing LM pectin, despite presenting a reduction in the contact angle, present an increase in the fiber diameter, ranging from 8.08 to 8.27 μm, with an increase in the amount of pectin (Figure 13). This behavior can be explained by the fact that in addition to the influence of the fiber diameter, the contact angle is influenced by the surface structure of the fibers and the composition of the fiber material. However, it can be seen that fibers containing HM pectin are more hydrophobic (Figure 15b) and present a reduction in the fiber diameter (Figure 14) with a reduction in the contact angle, following the trend of a smaller fiber diameter resulting in a smaller contact angle. The boost in hydrophobicity can be attributed to the morphological defect on the nanofibers, which likely enhances the surface roughness and consequently reduces the contact area between water droplets and the material (Figure 14d–f).

The TGA curves of the films modified with LM pectin powder and 0.3% PEO (a) or with 1% PEO (b) and with HM pectin powder with 0.3% PEO (c) or with 1% (d) showed that weight loss occurred in three distinct stages, as shown in Figure 16. The first stage was the release of free water up to 100 °C, followed by the evaporation of bound water in the range between 100 and 180 °C. The third stage was the degradation of the polymeric chains above 180 °C for both pectins [46].

The highest weight loss was observed in the 180 to 290 °C range, 65% for LM pectin powder, and 50% for HM pectin powder. The addition of pectin delayed the degradation of the composite films, as the onset temperature of the main degradation stage was higher than that of films without pectin. This improvement in thermal stability was more pronounced with the addition of both pectins.

Although the HM pectin powder exhibited a higher percentage of degradation at temperatures up to 250 °C, its addition to the films did not impair their thermal stability, making them comparable to the films with LM pectin powder.

These results were confirmed using FTIR and contact angle tests, which evidenced the proper dispersion of the pectin additive in the zein matrix. Therefore, the adequate dispersion of the pectin additive in the zein matrix effectively hindered the diffusion of volatile decomposition products and provided better thermal stability to the modified films [47].

The polysaccharide fibers of zein/carrageenan (90:10)/PEO1 (Z-C (90:10)/P1 sample), although they did not have the largest fiber diameter among the polysaccharides studied, show promise for application as fibers in plant-based meats. This is due to the excellent spinnability of the solution, the formation of continuous and homogeneous fibers, and their hydrophilic surface. These factors can contribute to improving the texture and appearance of plant-based meats. Therefore, using carrageenan as fibers in these meats can contribute to developing healthier and tastier products, meeting the demands of consumers looking for plant-based alternatives in their diet.

## 4. Conclusions

The electrospinning of polysaccharides is accompanied by significant challenges, as these polymers’ complex and viscous nature makes transforming them into fibers difficult. However, combining polysaccharides (alginate, carrageenan, or pectin) with zein and PEO has shown promising results. The Z-C (90:10)/P1 sample, in particular, stands out for its homogeneous and continuous structure, offering higher hydrophilicity than other polysaccharide samples. These attributes make it an ideal option for simulating fibers in plant-based meats, allowing for improved moisture retention and a more appealing texture. Notably, the zein/carrageenan/PEO fibers demonstrate resilience to high temperatures, retaining their physical properties, which is crucial for achieving a consistent and pleasant texture. Therefore, using carrageenan in fiber production for plant-based meats can produce high-quality products with sensory appeal and texture akin to traditional meats. By optimizing electrospinning conditions and polymer selection, producing high-quality fibers resembling those found in plant-based meats is achievable, catering to the increasing demand for vegan alternatives.

## Figures and Tables

**Figure 1 foods-13-02962-f001:**
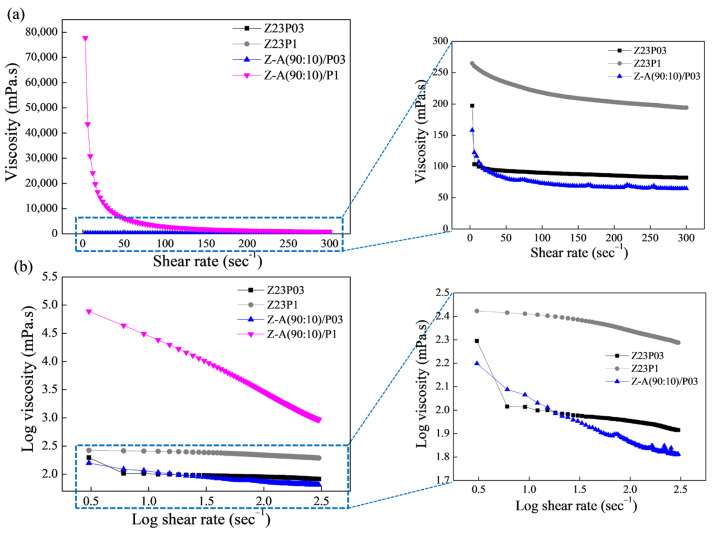
Viscosity as a function of shear rate (**a**) and log–log plot of the viscosity as a function of shear rate (**b**) for zein (23 wt.%)/PEO (0.3 wt.%), zein (23 wt.%)/PEO (1 wt.%), zein/alginate (90:10)/PEO 0.3%, and zein/alginate (90:10)/PEO 1% polymer solutions.

**Figure 2 foods-13-02962-f002:**
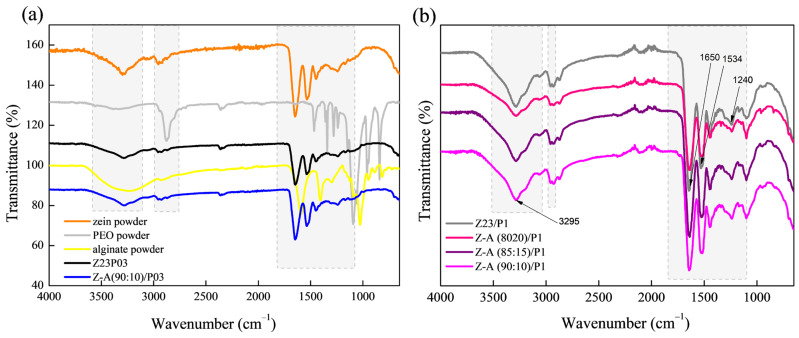
ATR-FTIR spectra of zein powder, PEO powder, alginate powder, Z23P03, and Z-A/(90:10)/P03) (**a**); and Z23P1, Z-A/(80:20)/P1, Z-A/(85:15)/P1, and Z-A/(90:10)/P1 (**b**).

**Figure 3 foods-13-02962-f003:**
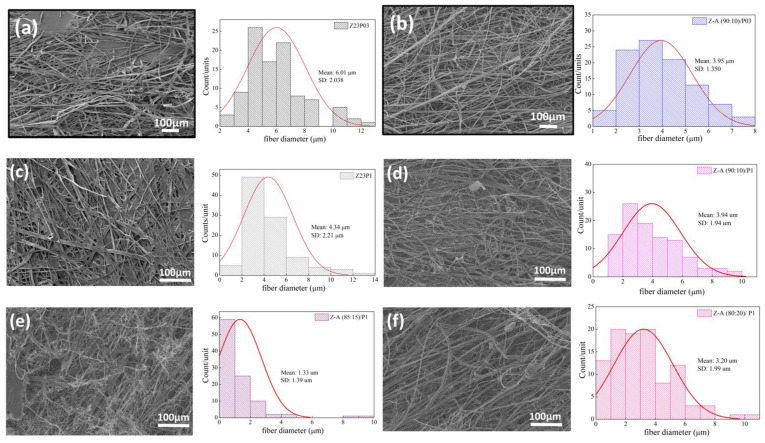
SEM images of Z23P03 (**a**), Z-A (90:10)/P03 (**b**), Z23P1 (**c**) Z-A (90:10)/P1 (**d**), Z-A (85:15)/P1 (**e**), and Z-A (80:20)/P1 (**f**) and their respective fiber diameters.

**Figure 4 foods-13-02962-f004:**
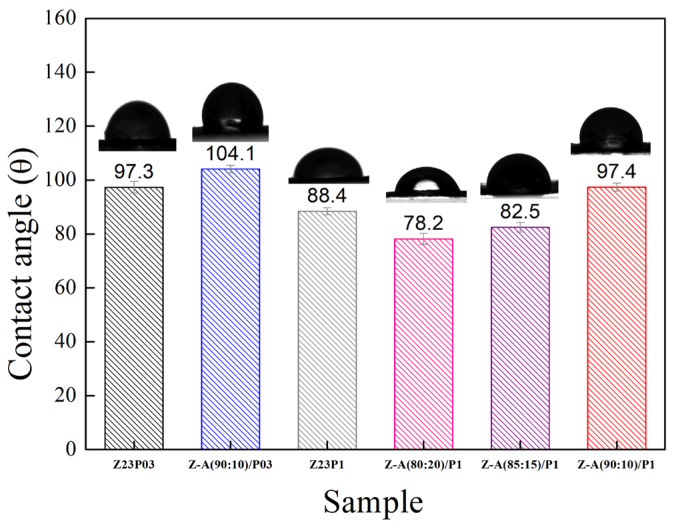
The contact angles of Z23P03, Z23P1, Z-A (90:10)/P03, Z-A (80:20)/P-1, Z-A (85:15)/P1, and Z-A (90:10)/P1 fibers.

**Figure 5 foods-13-02962-f005:**
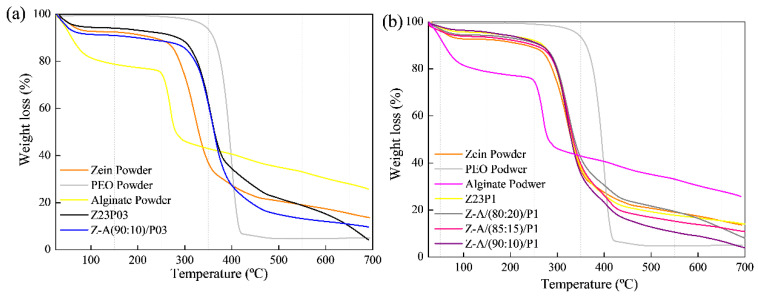
TG analysis of zein powder, PEO powder, alginate powder, Z23P03, Z-A/(80:20)/P03, Z-A/(85:15)/P03, and Z-A/(90:10)/P03 (**a**); TG analysis of zein powder, PEO powder, alginate powder, Z23P1, Z-A/(80:20)/P1, Z-A/(85:15)/P1, and Z-A/(90:10)/P1(**b**).

**Figure 6 foods-13-02962-f006:**
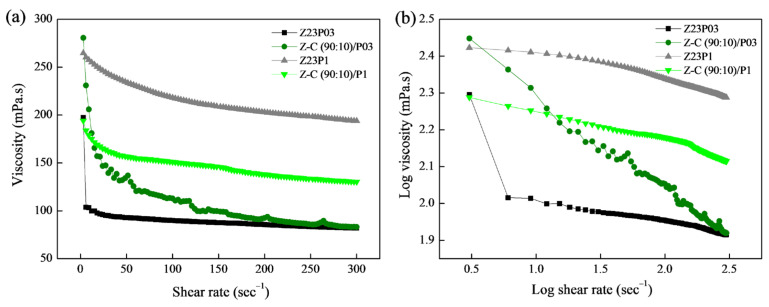
Viscosity as a function of shear rate (**a**) and log–log plot of the viscosity as a function of shear rate (**b**) for zein/PEO and zein/carrageenan/PEO solutions at different PEO concentrations.

**Figure 7 foods-13-02962-f007:**
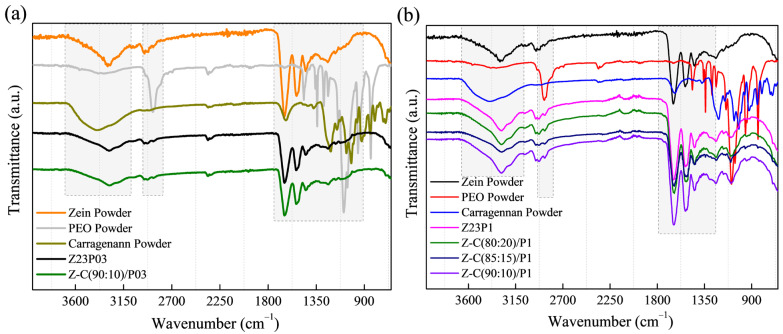
ATR-FTIR spectra of zein powder, PEO powder, carrageenan powder, Z23P03, and Z-C/(90:10)/P03 (**a**); and zein powder, PEO powder, carrageenan powder, Z23P1, Z-C/(80:20)/P1, Z-C/(85:15)/P1, and Z-C/(90:10)/P19 (**b**).

**Figure 8 foods-13-02962-f008:**
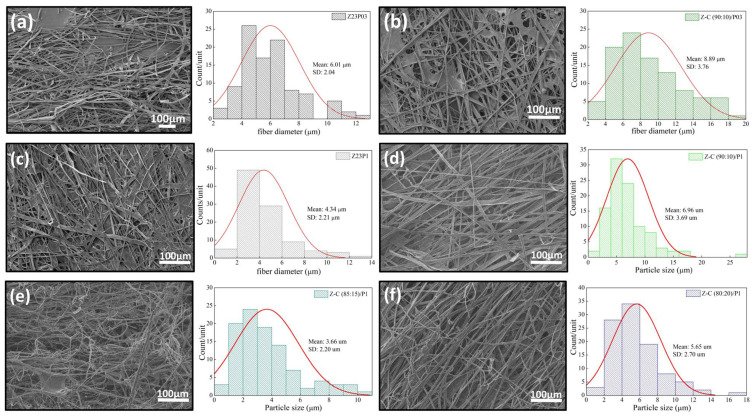
SEM images of Z23P03 (**a**), Z-C (90:10)/P03 (**b**), Z23P1 (**c**), Z-C (90:10)/P1 (**d**), Z-C (85:15)/P1 (**e**), and Z-C (80:20)/P1 (**f**) fibers and their respective fiber diameters.

**Figure 9 foods-13-02962-f009:**
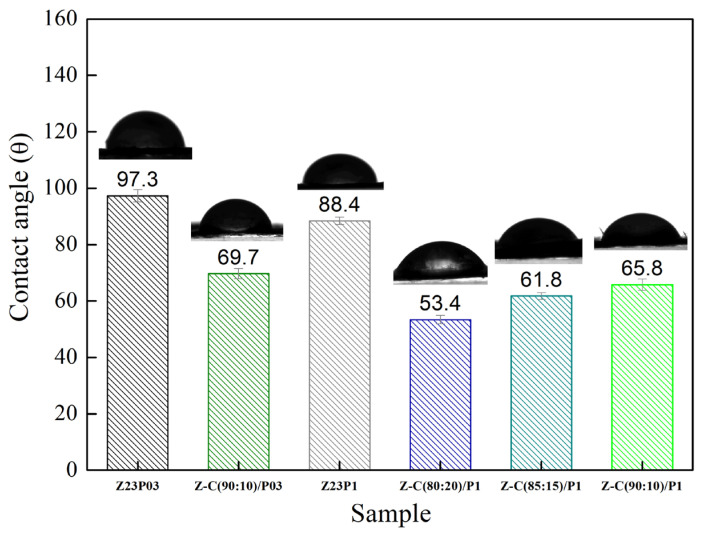
Contact angles of Z23P03, Z23P1, Z-C(90:10)/P03, Z-C(80:20)/P1, Z-C(85:15)/P1, and Z-C (90:10)/P1 fibers.

**Figure 10 foods-13-02962-f010:**
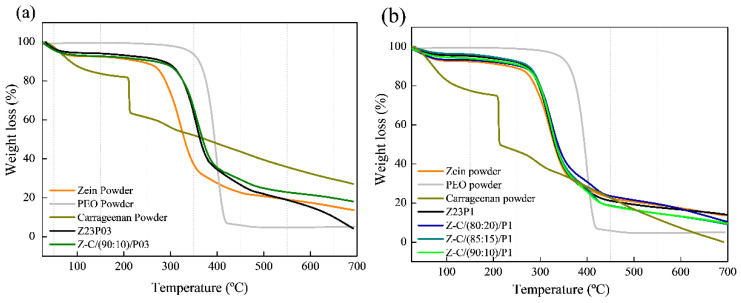
TG analysis of zein powder, PEO powder, carrageenan powder, Z23P03, and Z-C(90:10)/P03 (**a**) and zein powder, PEO powder, carrageenan powder, Z23P1, Z-C(80:20)/P1, Z-C(85:15)/P1, and Z-C(90:10)/P1(**b**).

**Figure 11 foods-13-02962-f011:**
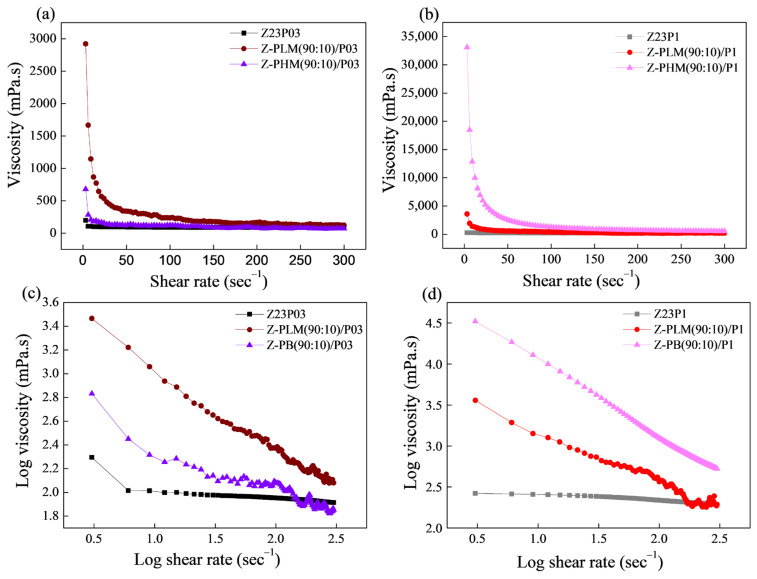
Viscosity as a function of shear rate (**a**,**b**) and log–log plot of the viscosity as a function of shear rate (**c**,**d**) for zein/PEO and zein/pectin/PEO solutions at different PEO concentrations.

**Figure 12 foods-13-02962-f012:**
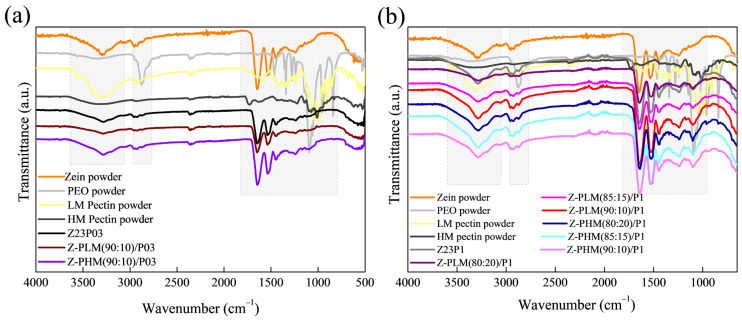
ATR-FTIR spectra of zein powder, PEO powder, LM pectin powder, Z23P03, and Z-PLM (90:10)/P03 (**a**); zein powder, PEO powder, LM pectin powder, Z23P1, Z-PLM (80:20)/P1, Z-PLM (85:15)/P1, and Z-PLM (90:10)/P1 (**b**); zein powder, PEO powder, LM pectin powder, Z23P03, and Z-PB (90:10)/P03; and zein powder, PEO powder, LM pectin powder, Z23P03, Z-PB (80:20)/P1, Z-PB (80:20)/P1, Z-PB (85:15)/P1, and Z-PB (90:10)/P1.

**Figure 13 foods-13-02962-f013:**
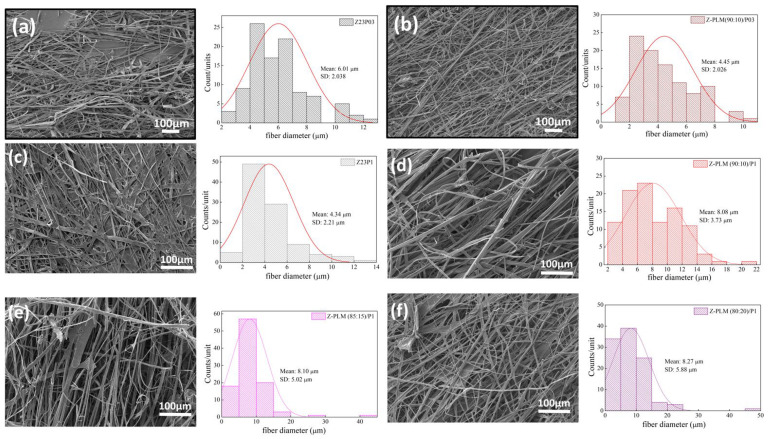
SEM images of Z23P03 (**a**), Z-PLM (90:10)/P03 (**b**), Z23P1 (**c**), Z-PLM (90:10)/P1 (**d**), Z-PLM (85:15)/P1 (**e**), and Z-PLM (80:20)/P1 (**f**) fibers and their respective fiber diameters.

**Figure 14 foods-13-02962-f014:**
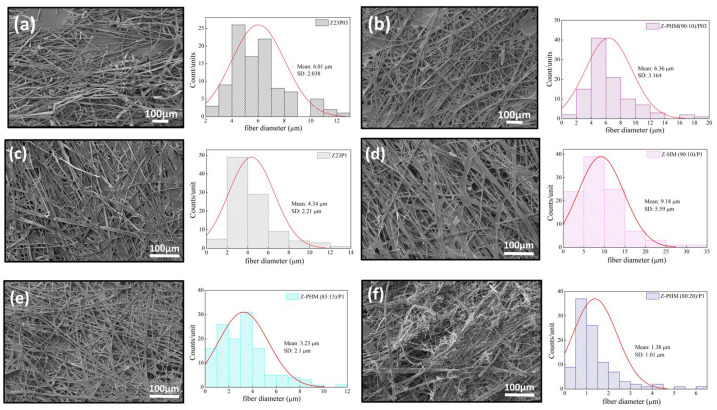
SEM images of Z23P03 (**a**), Z-PHM (90:10)/P03 (**b**), Z23P1 (**c**), Z-HM (90:10)/P1 (**d**), Z-PHM (85:15)/P1 (**e**), and Z-PHM (80:20)/P1 (**f**) fibers and their respective fiber diameters.

**Figure 15 foods-13-02962-f015:**
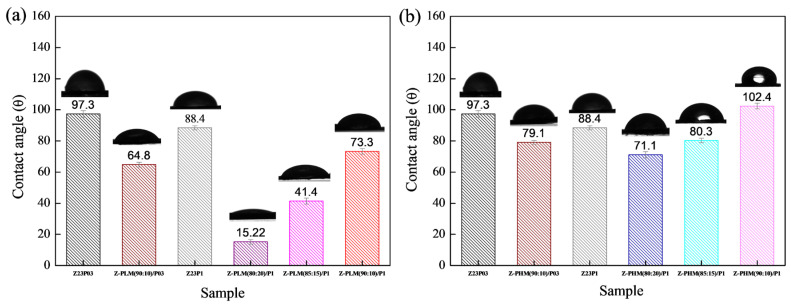
Contact angle of Z23P03, Z23P1, Z-PLM (90:10)/P03, Z-PLM (80:20)/P1, Z-PLM (85:15)/P1, and Z-PLM (90:10)/P1 (**a**) and Z23P03, Z23P1, Z-PHM (90:10)/P03, Z-PHM (80:20)/P1, Z-PHM (85:15)/P1, and Z-PHM (90:10)/P1 (**b**) fibers.

**Figure 16 foods-13-02962-f016:**
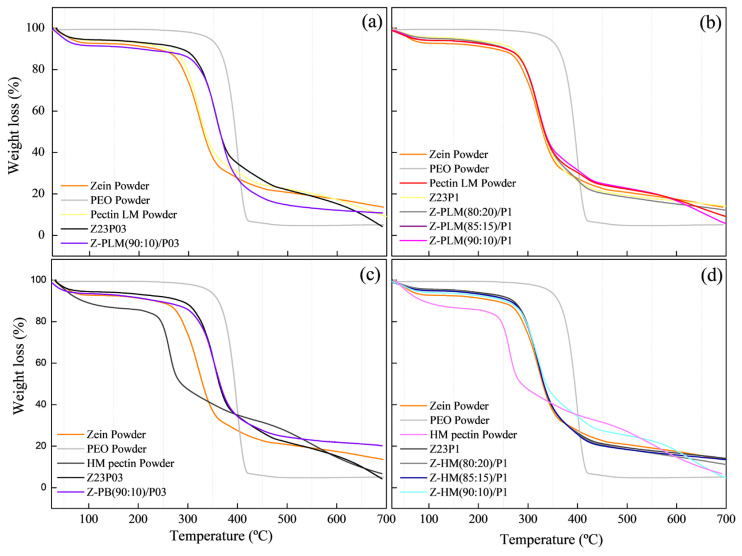
TG analysis of zein powder, PEO powder, LM pectin powder, Z23P03, and Z-PLM(90:10)/P03 (**a**); zein powder, PEO powder, LM pectin powder, Z23P03, Z-PLM/(80:20)P1, Z-PLM(85:15)/P1, and Z-PLM(90:10)/P1 (**b**); zein powder, PEO powder, HM pectin powder, Z23P03, and Z-PB(90:10)/P03 (**c**); and zein powder, PEO powder, HM pectin powder, Z23P1, Z-PB/(80:20)/P1, Z-PHM(85:15)/P1, and Z-PHM(90:10)/P1 (**d**).

**Table 1 foods-13-02962-t001:** The processing conditions and macroscopic and microscopic observations during and after the spinning process for Alginate/PEO combination.

Composition	Voltage (kV)	Feeding Rate (μL/h)	TCD (cm)	Visual Observation at Needle	Microscopic Observation on Collector
Alginate (2 wt %)	18	1800	15	Drops	None
Alginate (2 wt %)	22	2000	15	Drops	None
Alginate/PEO (20:80)	18	1800	15	Drops	None
Alginate/PEO (20:80)	22	2000	15	Drops	None
Alginate/PEO (40:60)	18	1800	15	Drops	None
Alginate/PEO (40:60)	22	2000	15	Drops	None
Alginate/PEO (50:50)	18	1800	15	Drops	None
Alginate/PEO (50:50)	22	2000	15	Drops	None

**Table 2 foods-13-02962-t002:** The processing conditions and macroscopic and microscopic observations during and after the spinning process for Zein/Alginate/PEO combination.

Composition	Sample	Voltage (kV)	Feeding Rate (μL/h)	TCD (cm)	Visual Observation at Needle	Microscopic Observation on Collector
Zein 23%/PEO 0.3%	Z23P03	24	2000	12	Jet	Fibers
Zein/alginate (80:20)/PEO 0.3%	Z-A/(80:20)/P03	22	2000	15	Jet and drops	None
Zein/alginate (80:20)/PEO 0.3%	Z-A/(80:20)/P03	24	2000	12	Jet and drops	None
Zein/alginate (85:15)/PEO 0.3%	Z-A/(85:15)/P03	22	2000	15	Jet and drops	None
Zein/alginate (85:15)/PEO 0.3%	Z-A/(85:15)/P03	24	2000	12	Jet and drops	None
Zein/alginate (90:10)/PEO 0.3%	Z-A/(90:10)/P03	22	2000	15	Jet and drops	None
Zein/alginate (90:10)/PEO 0.3%	Z-A/(90:10)/P03	24	2000	12	Jet	Fibers
Zein 23%/PEO 1%	Z23P1	24	2000	12	Jet	Fibers
Zein/alginate (80:20)/PEO 1%	Z-A/(80:20)/P1	24	2000	12	Jet	Fibers
Zein/alginate (85:15)/PEO 1%	Z-A/(85:15)/P1	24	2000	12	Jet	Fibers
Zein/alginate (90:10)/PEO 1%	Z-A/(90:10)/P1	24	2000	12	Jet	Fibers

**Table 3 foods-13-02962-t003:** Power Law model for the rheological parameters and standard deviation of polymer solutions.

Parameter	Z23P03	Z23P1	Z-A (90:10)/P 03	Z-A (90:10)/P 1
k (Pa·s^n^)	0.120 ^c^ ± 0.002	0.383 ^b^ ± 0.008	0.140 ^c^ ± 0.130	16.20 ^a^ ± 2.633
n (−)	0.932 ^a^ ± 0.0	0.880 ^b^ ± 0.0	0.876 ^b^± 0.13	0.490 ^c^ ± 0.03
R^2^	99.99	99.93	99.81	99.80
Viscosity 10 s^−1^ (mPa·s)	95.6 ^c^ ± 1.70	252.8 ^b^ ± 2.83	143.5 ^c^ ± 70.19	5421.2 ^a^ ± 880.99

Different letters indicate a significant difference (*p* < 0.05) in the same line. Z23P03: Zein 23%/PEO 0.3%, Z23P1: Zein 23%/PEO 1%, Z-A (90:10)/P03: Zein/alginate (90:10)/PEO 0.3%, and Z-A (90:10)/P01: Zein/alginate (90:10)/PEO 1%

**Table 4 foods-13-02962-t004:** The macroscopic and microscopic observations during and after the spinning process and the Power Law parameters for the polymer solutions containing Zein/Carrageenan/PEO

Composition	Sample	Visual Observation at Needle	Microscopic Observation on Collector	k (Pa·s^n^)	n (-)	R^2^	Viscosity 10 s^−1^ (mPa·s)
Zein 23%/PEO 0.3%	Z23P03	Jet	Fibers	0.120 ^d^ ± 0.002	0.932 ^a^ ± 0.0	99.99	95.58 ^d^ ± 1.70
Zein/carrageenan (80:20)/PEO 0.3%	Z-C(80:20)/P03	Jet and drops	None	-	-	-	-
Zein/carrageenan (85:15)/PEO 0.3%	Z-C(85:15)/P03	Jet and drops	None	-	-	-	-
Zein/carrageenan (90:10)/PEO 0.3%	Z-C(90:10)/P03	Jet	Fibers	0.355 ^b^ ± 0.013	0.745 ^d^ ± 0.01	99.88	206.0 ^b^ ± 13.56
Zein 23%/PEO 1%	Z23P1	Jet	Fibers	0.383 ^a^ ± 0.008	0.880 ^b^ ± 0.0	99.93	252.8 ^a^ ± 2.83
Zein/carrageenan (80:20)/PEO 1%	Z-C(80:20)/P1	Jet	Fibers	-	-	-	-
Zein/carrageenan (85:15)/PEO 1%	Z-C(85:15)/P1	Jet	Fibers	-	-	-	-
Zein/carrageenan (90:10)/PEO 1%	Z-C(90:10)/P1	Jet	Fibers	0.272 ^c^ ± 0.014	0.872 ^c^ ± 0.01	99.95	178.8 ^c^ ± 6.83

Different letters indicate a significant difference (*p* < 0.05) in the same column.

**Table 5 foods-13-02962-t005:** The macroscopic and microscopic observations during and after the spinning process and the Power Law parameters for the polymer solutions containing Zein/PEO or Zein/pectin/PEO

Composition	Voltage (kV)	Visual Observation at Needle	Microscopic Observation on Collector	k (Pa·s^n^)	n (-)	R^2^	Viscosity 10 s^−1^ (mPa·s)
Zein 23%/PEO 0.3%	Z23P03	Jet	Fibers	0.120 ^d^ ± 0.002	0.932 ^a^ ± 0.0	99.99	95.58 ^d^ ± 1.70
Zein/HM pectin (80:20)/PEO 0.3%	Z-PHM(80:20)/P03	Jet and drops	None	-	-	-	-
Zein/HM pectin (85:15)/PEO 0.3%	Z-PHM(85:15)/P03	Jet and drops	None	-	-	-	-
Zein/HM pectin (90:10)/PEO 0.3%	Z-PHM(90:10)/P03	Jet	Fibers	0.85761 ^b^ ± 0.581	0.573 ^d^ ± 0.11	95.91	394.1 ^c^ ± 80.37
Zein/LM pectin (80:20)/PEO 0.3%	Z-PLM(80:20)/P03	Jet and drops	None	-	-	-	-
Zein/LM pectin (85:15)/PEO 0.3%	Z-PLM(85:15)/P03	Jet and drops	None	-	-	-	-
Zein/LM pectin (90:10)/PEO 0.3%	Z-PLM(90:10)/P03	Jet	Fibers	-	-	-	2021.0 ^a^ ± 977.38
Zein 23%/PEO 1%	Z23P1	Jet	Fibers	0.383 ^c^ ± 0.008	0.880 ^b^ ± 0.0	99.93	252.8 ^c^ ± 2.83
Zein/HM pectin (80:20)/PEO 1%	Z-PHM(80:20)/P1	Jet	Fibers	-	-	-	-
Zein/HM pectin (85:15)/PEO 1%	Z-PHM(85:15)/P1	Jet	Fibers	-	-	-	-
Zein/HM pectin (90:10)/PEO 1%	Z-PHM(90:10)/P1	Jet	Fibers	3.82 ^a^ ± 0.327	0.658 ^c^ ± 0.02	99.97	1742.6 ^a^ ± 101.76
Zein/LM pectin (80:20)/PEO 1%	Z-PLM(80:20)/P1	Jet	Fibers	-	-	-	-
Zein/LM pectin (85:15)/PEO 1%	Z-PLM(85:15)/P1	Jet	Fibers	-	-	-	-
Zein/LM pectin (90:10)/PEO 1%	Z-PLM(90:10)/P1	Jet	Fibers	4.90 ^a^ ± 1.114	0.399 ^e^ ± 0.06	86.13	991.5 ^b^ ± 52.09

Different letters indicate a significant difference (*p* < 0.05) in the same column.

## Data Availability

The original contributions presented in the study are included in the article, further inquiries can be directed to the corresponding author.

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
