# Peer review of "Spinning a Sustainable Future: Electrospun Polysaccharide–Protein Fibers for Plant-Based Meat Innovation"

_foods, 2024, doi:10.3390/foods13182962_

Round 1
Reviewer 1 Report
Comments and Suggestions for Authors
Reviewer Comments
Extensive work was done on the manuscript entitled "Spinning a Sustainable Future: Electrospun Polysaccharide-Protein Fibers for Plant-Based Meat Innovation". The manuscript evaluates the feasibility of producing electrospun fibers by combining polysaccharides, zein, and poly(ethylene oxide) (PEO) to simulate fibers applied in plant-based meat analogs. This is indeed a very interesting question, and by optimising electrospinning conditions and polymer selection, it is possible to produce high-quality fibres similar to plant meat to meet the growing demand for vegan alternatives.
1. Line 47, "Finally" is not appropriate here.
2. The foreword should be supplemented with the current status of research by scholars.
3. In "2.4. Characterisation of electrospun fibers", the country information of the instrument should be added.
4. The title "3.1 XXX" is missing in the manuscript.
5. Lines 219-220, please explain the reason for the decrease in the number of fibres.
6. For the analysis of the SEM results, the discussion of the results and the analysis of the reasons should be supplemented, for example, why the lower the concentration of carrageenan, the larger and more uniform the fibres.
7. Lines 315-317, please explain the reason for the increase in film degradation temperature.

Comments on the Quality of English LanguageMinor modifications to the English language are required.
Author Response
Responses to Technical Check Results
Santos, August 31st, 2024
Manuscript ID: foods-3182093
Title: Spinning a Sustainable Future: Electrospun Polysaccharide-Protein Fibers for Plant-Based Meat Innovation
Dear Editor,
We thank the reviewers for their valuable comments and suggestions on our manuscript. We have modified it according to the recommendations, and the detailed, point-by-point corrections are listed below. The modifications requested were extremely important to improve our work. All sentences/words listed below are highlighted in the revised version of the manuscript.
Reviewer 1
Comment: Extensive work was done on the manuscript entitled "Spinning a Sustainable Future: Electrospun Polysaccharide-Protein Fibers for Plant-Based Meat Innovation". The manuscript evaluates the feasibility of producing electrospun fibers by combining polysaccharides, zein, and poly(ethylene oxide) (PEO) to simulate fibers applied in plant-based meat analogs. This is indeed a very interesting question, and by optimising electrospinning conditions and polymer selection, it is possible to produce high-quality fibres similar to plant meat to meet the growing demand for vegan alternatives.
Response: The authors appreciate the valuable suggestions.
Comment: Line 47, "Finally" is not appropriate here.
Response: The word was replaced.
Comment: The foreword should be supplemented with the current status of research by scholars.
Response: The information was added.
Comment: In "2.4. Characterisation of electrospun fibers", the country information of the instrument should be added.
Response: The information was added.
Comment: The title "3.1 XXX" is missing in the manuscript.
Response: The requested information was added.
Comment: Lines 219-220, please explain the reason for the decrease in the number of fibres.
Response: The requested information was added.
Comment: For the analysis of the SEM results, the discussion of the results and the analysis of the reasons should be supplemented, for example, why the lower the concentration of carrageenan, the larger and more uniform the fibres.
Response: The requested information was added.
Comment: Lines 315-317, please explain the reason for the increase in film degradation temperature.
Response: We appreciate your comment. However, as we wrote in the text, the addition of carrageenan does not interfere with the thermal stability of the fibers compared to zein/PEO fibers. There is an increase in degradation when we compare both zein/PEO fibers and zein/carrageenan/PEO fibers with the degradation of PEO in powder form. And an increase in thermal stability when we compare the degradation of these fibers with the degradation of zein in powder form.
Reviewer 2 Report
Comments and Suggestions for Authors
Dear researchers, here you are with some questions and remarks for the manuscript:
Abstract:
Is poly(ethylene oxide) (PEO) allowed to be used in food?
Could you please clarify the percentage mentioned in the abstract cit. „The results indicated that the electrospun fibers containing 23% zein-1% carrageenan 16 (mixed at proportion 90:10)/ 1% PEO presented a promising...“
1. Methodology:
1.1. Make more clear (and therefore easy understandable) the content of the subsection 2.1.1. Alginate (Line 115), see the recommendation below:
The subsection 2.1.1, titled "Alginate," should be divided into two parts (sections) with distinct headings. The first section with the different title should focus exclusively on the preparation of the alginate and PEO mixture, while the second part should address the preparation of the ZEIN/alginate/PEO solution. For example, the subsequent subsection 2.1.2, titled "Carrageenan," only addresses the production of the ZEIN/Carrageenan/PEO mixture. This recommendation is suggested to maintain consistency throughout the text. Please rename sections 2.1.1, 2.1.2, and 2.1.3 to enhance their logical coherence and content consistency.
1.2. Please specify the final ratios of ZEIN/alginate that were chosen. Ensure that these ratios are presented in the same order (ZEIN/alginate/PEO) as in subsections 2.1.2 and 2.1.3., see subsection 2.1.1. Alginate (Lines 120-125).
1.3. Can you elaborate on why specific proportions of zein, carrageenan, and PEO were chosen for the electrospinning process? Was this based on previous research or specific functional properties? How do these ratios affect the overall performance of the fibers?
1.4. Could you please provide more detailed reasoning behind the selected electrospinning parameters (voltage, flow rate, and solution concentration)? Were these parameters optimized for the specific materials used, and if so, what were the criteria for this optimization?
1.5. There is no statistical evaluation method description provided in the manuscript. Please include details about the sample sizes, number of replicates, and the statistical methods used in the analysis of your results in a separate section dedicated to the statistical evaluation methodology. Additionally, can you clarify if the differences in rheological and contact angle measurements were statistically significant?
2.Results
2.1. Could you please explain, why were only the voltage and TCD parameters adjusted during electrospinning? Was the environmental humidity content also considered? see section 3 Results (Lines 203-204).
2.2. Could you please provide and interpret the contact angle data from the experimental results in Figure 4 and Figure 15? Are there other factors (such as fiber density or porosity) that might influence the performance of these fibers in plant-based meat applications beyond the contact angle measurements?
2.3. Table data, Section 3 Results Table 2 (Line 211) and Table 4 (Line 355)
The data in the table appears to be misaligned, with entries not matching their corresponding columns. The table's format needs to be adjusted so that the data is correctly placed under the appropriate column headings.
2.4. Section 3 Results Line 214, please explain, why was the environmental humidity content not evaluated alongside the other parameters?
2.5. Subsection 3.1 is missing. Could you please include it in the content or ensure it is included?
2.6. Application of fibers in food systems:
While the fibers are suggested to be suitable for use in plant-based meat analogs, can you share insights or data on how they would perform in real-world food applications? Have you considered factors like texture, stability, or consumer acceptability, and how do the fibers compare to current alternatives used in the industry?
Line 18: Instead of "because presented a promising structure" suggest: "because they presented a promising structure"
Line 21: Instead of "helps the spinnability", suggest: "helps with the spinnability"
Line 100: Instead of "create new or improved textural attributes in foods" suggest „create new or improved textures in foods"
Line 132: Instead of "Concomitantly, a solution of 23 wt.% zein in 80% ethanol was prepared and kept under stirring for 1 h" suggest: "Simultaneously, a 23 wt.% zein solution in 80% ethanol was prepared and stirred for 1 hour".
Line 147: Instead of "continued for another 1 h" suggest: "continued for another hour".
Line 432: Please correct the title of table 5 „Table 5. This is a table. Tables should be placed in the main text near the first time they are cited“.
The quality of Figures 3, 8, 13, and 14 should be improved, as the data presented in these figures is not clearly visible.
The labeling of the X-axis in graphs a and b of Figures 3, 8, 13, and 14 differs from the X-axis labels in graphs c-f of the same figures. Is this difference correct, considering that the morphology part only discusses fiber diameters (as named in X axis of a-b Figures) and not particle sizes (as named in X axis of c-f Figures)?
Comments on the Quality of English Languagenone
Author Response
Responses to Technical Check Results
Santos, August 31st, 2024
Manuscript ID: foods-3182093
Title: Spinning a Sustainable Future: Electrospun Polysaccharide-Protein Fibers for Plant-Based Meat Innovation
Dear Editor,
We thank the reviewers for their valuable comments and suggestions on our manuscript. We have modified it according to the recommendations, and the detailed, point-by-point corrections are listed below. The modifications requested were extremely important to improve our work. All sentences/words listed below are highlighted in the revised version of the manuscript.
Reviewer 2
Comment: Dear researchers, here you are with some questions and remarks for the manuscript. Abstract: Is poly(ethylene oxide) (PEO) allowed to be used in food?
Response: Adding small quantities of poly(ethylene oxide) (PEO) is an alternative to provide elasticity to the formation of fibers combined with zein and other proteins. PEO is a hydrophilic polymer safely used in food fields due to its non-toxicity, biocompatibility, and biodegradability. Moreover, among the polymers reported in the literature, PEO is biodegradable, has a high molecular weight, and is certified as Generally Recognized as Safe (GRAS) (FDA UNII 16P9295IIL) and has been approved by the Food and Drug Administration (FDA). Hence, it can be safely used in processed food and beverages.
Comment: Abstract - Could you please clarify the percentage mentioned in the abstract cit. „The results indicated that the electrospun fibers containing 23% zein-1% carrageenan 16 (mixed at proportion 90:10)/ 1% PEO presented a promising...“
Response: Thank you for the comments. The results indicated that the fibers prepared in the proportions 90:10 of zein/carrageenan from the mixture of a solution containing 23 wt.% of zein with a solution containing 1 wt.% of carrageenan and with the addition of 1 wt.% of PEO presented a promising structure for application as fibers in meat analogs because they have a more hydrophilic surface.
Comment: Make more clear (and therefore easy understandable) the content of the subsection 2.1.1. Alginate (Line 115), see the recommendation below:
The subsection 2.1.1, titled "Alginate," should be divided into two parts (sections) with distinct headings. The first section with the different title should focus exclusively on the preparation of the alginate and PEO mixture, while the second part should address the preparation of the ZEIN/alginate/PEO solution. For example, the subsequent subsection 2.1.2, titled "Carrageenan," only addresses the production of the ZEIN/Carrageenan/PEO mixture. This recommendation is suggested to maintain consistency throughout the text. Please rename sections 2.1.1, 2.1.2, and 2.1.3 to enhance their logical coherence and content consistency.
Response: We appreciate your comment. The indicated sections have been reorganized.
Comment: Please specify the final ratios of ZEIN/alginate that were chosen. Ensure that these ratios are presented in the same order (ZEIN/alginate/PEO) as in subsections 2.1.2 and 2.1.3., see subsection 2.1.1. Alginate (Lines 120-125).
Response: We appreciate your comment. The final ratios have been specified.
Comment: Can you elaborate on why specific proportions of zein, carrageenan, and PEO were chosen for the electrospinning process? Was this based on previous research or specific functional properties? How do these ratios affect the overall performance of the fibers?
Response: We appreciate your comment. The specific proportions of zein/carrageenan/PEO as well as the zein/pectin/PEO proportions were chosen for the electrospinning process based on the results obtained for alginate. Thus, maintaining the same zein/polysaccharide/PEO proportions. The results obtained showed that the higher the zein concentration, the more electrospinnable the polymer solution is.
Comment: Could you please provide more detailed reasoning behind the selected electrospinning parameters (voltage, flow rate, and solution concentration)? Were these parameters optimized for the specific materials used, and if so, what were the criteria for this optimization?
Response: We appreciate your comment. The electrospinning process is influenced by several factors that also influence the properties of the obtained fibers. The parameters tested in this work for the electrospinning of solutions with polysaccharides were based on the literature [https://doi.org/10.3390/pr10122737]. Based on the parameters presented in the literature, we adjusted these parameters for our zein/alginate/PEO solutions (Table 2), concluding that the best parameters were voltage of 24 kV, flow rate of 2000 μL/h and TCD of 12 cm. These parameters were maintained for the solutions containing carrageenan and pectin. As for the concentration of the solution, this was varied since the results showed that a solution with 2 wt.% of Alginate as well as the addition of PEO to the alginate solution (Table 1) did not allow obtaining fibers. We therefore sought an alternative to electrospin a solution with polysaccharide. The solution found was to make a blend with zein and to improve reliability, we added PEO. Since the polysaccharide solutions proved to be very viscous, and the viscosity of the polymer solution affects the formation of the jet, we chose to start with a 90:10 zein to alginate ratio to favor the electrospinnability of the solution, reaching a 80:20 zein:alginate ratio with 1 wt.% PEO to obtain fibers. Thus, the parameters successfully tested for alginate were maintained for the other polysaccharides studied.
Comment: There is no statistical evaluation method description provided in the manuscript. Please include details about the sample sizes, number of replicates, and the statistical methods used in the analysis of your results in a separate section dedicated to the statistical evaluation methodology. Additionally, can you clarify if the differences in rheological and contact angle measurements were statistically significant?
Response: Statistical evaluation method description was provided.
Comment: Could you please explain, why were only the voltage and TCD parameters adjusted during electrospinning? Was the environmental humidity content also considered? see section 3 Results (Lines 203-204).
Response: We appreciate your comment. In addition to the voltage and TCD parameters, we adjusted the viscosity of the solutions, as shown in the study of the electrospinnability of alginate, Tables 1 and 2. From the best ratio found for alginate, we fixed it for the other polysaccharides studied. And we also evaluated the feeding rate (μL/h) for alginate, Table 1. So, we fixed the feeding rate (μL/h) at 2000 μL/h. As written in section "2.3. Preparation of polysaccharides/protein fibers" all nanofibers were made at room temperature (20-25 °C) and 50–60% relative humidity.
Comment: Could you please provide and interpret the contact angle data from the experimental results in Figure 4 and Figure 15? Are there other factors (such as fiber density or porosity) that might influence the performance of these fibers in plant-based meat applications beyond the contact angle measurements?
Response: We appreciate your comment. The contact angle data were provided. Yes, there are a number of factors that can influence the performance of plant-based fibers in plant-based meat applications, in addition to contact angle. These include density, porosity, chemical composition, fiber size, solubility, interactions with other ingredients, the method of fiber processing (e.g., extrusion, cooking, drying), hydration and emulsification (the ability of fibers to retain water and form emulsions), and pH and temperature (which can affect fiber hydration and stability). All of these factors interact in complex ways, and the combination of these factors can have a major impact on the texture, flavor, and acceptance of plant-based meat products.
Comment: Table data, Section 3 Results Table 2 (Line 211) and Table 4 (Line 355). The data in the table appears to be misaligned, with entries not matching their corresponding columns. The table's format needs to be adjusted so that the data is correctly placed under the appropriate column headings.
Response: We appreciate your comment. Tables 2 and 4 have been corrected.
Comment: Section 3 Results Line 214, please explain, why was the environmental humidity content not evaluated alongside the other parameters?
Response: We appreciate your comment. As written in section "2.3. Preparation of polysaccharides/protein fibers" all nanofibers were made at room temperature (20-25 °C) and 50–60% relative humidity. Therefore, in this work, humidity had no relevance in the conditions for the formation of fibers.
Comment: Subsection 3.1 is missing. Could you please include it in the content or ensure it is included?
Response: The information was added.
Comment: Application of fibers in food systems: While the fibers are suggested to be suitable for use in plant-based meat analogs, can you share insights or data on how they would perform in real-world food applications? Have you considered factors like texture, stability, or consumer acceptability, and how do the fibers compare to current alternatives used in the industry?
Response: The present study was carried out with the aim of evaluating different polymers and their combinations to determine which of these combinations in different percentages would be possible to manufacture, with their additional characterization. From these data, other texture evaluations such as those recommended will be carried out with the most promising formulations. The volume of data generated to date has already been quite extensive, so the addition of new tests in this same manuscript would make the discussion of the data difficult.
Comment: Line 18: Instead of "because presented a promising structure" suggest: "because they presented a promising structure"
Response: The change was made as requested.
Comment: Line 21 Instead of "helps the spinnability", suggest: "helps with the spinnability"
Response: The change was made as requested.
Comment: Line 100: Instead of "create new or improved textural attributes in foods" suggest „create new or improved textures in foods"
Response: The change was made as requested.
Comment: Line 132: Instead of "Concomitantly, a solution of 23 wt.% zein in 80% ethanol was prepared and kept under stirring for 1 h" suggest: "Simultaneously, a 23 wt.% zein solution in 80% ethanol was prepared and stirred for 1 hour".
Response: The change was made as requested.
Comment: Line 147: Instead of "continued for another 1 h" suggest: "continued for another hour".
Response: The change was made as requested.
Comment: Line 432: Please correct the title of table 5 „Table 5. This is a table. Tables should be placed in the main text near the first time they are cited“.
Response: The Table title was corrected.
Comment: The quality of Figures 3, 8, 13, and 14 should be improved, as the data presented in these figures is not clearly visible.
Response: The change was made as requested.
Comment: The labeling of the X-axis in graphs a and b of Figures 3, 8, 13, and 14 differs from the X-axis labels in graphs c-f of the same figures. Is this difference correct, considering that the morphology part only discusses fiber diameters (as named in X axis of a-b Figures) and not particle sizes (as named in X axis of c-f Figures)?
Response: The change was made as requested.
Round 2
Reviewer 2 Report
Comments and Suggestions for Authors
Thank you for your detailed responses and improvements.
However, we were specifically looking for the contact angle data from Figures 4 and 15, as well as an interpretation of these values in the context of the experimental results. Could you kindly provide those specific numbers and discuss their significance?
Would you consider adjusting the environmental humidity during the electrospinning process to improve fiber formation in cases where only droplets were obtained instead of fibers?
Author Response
Responses to Technical Check Results
Santos, September 5th, 2024
Manuscript ID: foods-3182093
Title: Spinning a Sustainable Future: Electrospun Polysaccharide-Protein Fibers for Plant-Based Meat Innovation
Dear Editor,
We thank the reviewers for their valuable comments and suggestions on our manuscript. We have modified it according to the recommendations, and the detailed, point-by-point corrections are listed below. The modifications requested were extremely important to improve our work. All sentences/words listed below are highlighted in the revised version of the manuscript.
Reviewer 2
Comment: 1. However, we were specifically looking for the contact angle data from Figures 4 and 15, as well as an interpretation of these values in the context of the experimental results. Could you kindly provide those specific numbers and discuss their significance?
Response: The Figures were modified to add the requested information.
Comment: Would you consider adjusting the environmental humidity during the electrospinning process to improve fiber formation in cases where only droplets were obtained instead of fibers?
Response: In the electrospinning process, we worked with 50–60% relative humidity since the humidity level of the environment can have a significant impact on the morphology of the formed fibers. It is recommended to maintain the relative humidity of the environment at levels between 30%-50%, to maximize the production of fibers instead of droplets. We worked with a relatively low humidity level and for most of the formulations tested, we obtained fiber formation. Perhaps, the polymer solutions tested where there was no fiber formation would benefit from a drier environment. However, it is worth remembering that other factors such as the solution viscosity, the applied voltage, the distance between the needle and the collector, and the composition of the polymer solution also play crucial roles in fiber formation and must be considered along with humidity.